# Understanding 6th-century barbarian social organization and migration through paleogenomics

Carlos Eduardo G. Amorim [1], Stefania Vai [2], Cosimo Posth [3,4], Alessandra Modi [2], István Koncz[5], Susanne Hakenbeck [6], Maria Cristina La Rocca[7], Balazs Mende[8], Dean Bobo[1], Walter Pohl[9], Luisella Pejrani Baricco[10], Elena Bedini[10], Paolo Francalacci[11], Caterina Giostra[10], Tivadar Vida[5,8], Daniel Winger[12], Uta von Freeden[13], Silvia Ghirotto[14], Martina Lari[2], Guido Barbujani[14], Johannes Krause [3,4], David Caramelli[2], Patrick J. Geary[15] & Krishna R. Veeramah[1]

Despite centuries of research, much about the barbarian migrations that took place between the fourth and sixth centuries in Europe remains hotly debated. To better understand this key era that marks the dawn of modern European societies, we obtained ancient genomic DNA from 63 samples from two cemeteries (from Hungary and Northern Italy) that have been previously associated with the Longobards, a barbarian people that ruled large parts of Italy for over 200 years after invading from Pannonia in 568 CE. Our dense cemetery-based sampling revealed that each cemetery was primarily organized around one large pedigree, suggesting that biological relationships played an important role in these early medieval societies. Moreover, we identified genetic structure in each cemetery involving at least two groups with different ancestry that were very distinct in terms of their funerary customs. Finally, our data are consistent with the proposed long-distance migration from Pannonia to Northern Italy.

[1] Department of Ecology and Evolution, Stony Brook University, Stony Brook, NY 11790, USA. [2] Dipartimento di Biologia Università degli Studi di Firenze, 50122 Firenze, Italy. [3] Max Planck Institute for the Science of Human History, Kahlaische Straße 10, 07745 Jena, Germany. [4] Institute for Archaeological Sciences Archaeo- and Palaeogenetics, University of Tübingen, Rümelinstraße 23, 72070 Tübingen, Germany. [5] Institute of Archaeological Sciences, Eötvös Loránd University, Múzeum körút 4/B, Budapest 1088, Hungary. [6] Department of Archaeology, University of Cambridge, Cambridge CB2 3DZ, UK. [7] Dipartimento DISSGeA, Università degli studi di Padova, 35100 Padova, Italy. [8] Research Centre for the Humanities, Hungarian Academy of Sciences, Budapest, Hungary. [9] Institut für Mittelalterforschung, Österreichische Akademie der Wissenschaften, 1020 Vienna, Austria. [10] Dipartimento di Storia, Archeologia e Storia dell'Arte, Università cattolica del Sacro Cuore, 20123 Milano, Italy. [11] Dipartimento di Scienze della Vita e dell'Ambiente, Università di Cagliari, Via T. Fiorelli, 1, 09126 Cagliari, Italy. [12] Heinrich Schliemann-Institut für Altertumswissenschaften Universität Rostock, 18055 Rostock, Germany. [13] Römisch-Germanische Kommission des Deutschen Archäologischen Instituts, 60325 Frankfurt am Main, Germany. [14] Dipartimento di Scienze della Vita e Biotenologie, Università degli Studi di Ferrara, 44121 Ferrara, Italy. [15] Institute for Advanced Study, Princeton, NJ 08540, USA. These authors contributed equally: Carlos Eduardo G. Amorim, Stefania Vai, Cosimo Posth. Correspondence and requests for materials should be addressed to J.K. (email: krause@shh.mpg.de) or to D.C. (email: david.caramelli@unifi.it) or to P.J.G. (email: geary@ias.edu) or to K.R.V. (email: krishna.veeramah@stonybrook.edu)

estern Europe underwent a major socio-cultural and economic transformation from Late Antiquity to the Early Middle Ages (i.e., third to tenth centuries CE). This period is often characterized by two major events: the collapse of the Western Roman Empire, and its invasion by various western and eastern barbarian/non-Romanized peoples such as the Goths, Franks, Anglo-Saxons, and Vandals, as well as by nomadic Huns; as such it has come to be known as the Migration Period, in German the Völkerwanderung, and in French Les invasions barbares. However, written accounts of these events are laconic, stereotypical, and largely written decades or even centuries later[1,2]. Because barbarian populations of the Migration Period left no written record, the only direct evidence of their societies comes from their archeological remains, chiefly grave goods, that have been used to make inferences about group identities, social structures, and migration patterns[3–5]. Unfortunately, grave goods represent a limited and highly curated portion of material culture, and with little other archeological data available, fundamental questions about barbarian social organization and migration remain unanswered[6] and vigorously debated[7,8]. Were specific barbarian peoples described in texts culturally and ethnically homogeneous populations, or were they ad-hoc and opportunistic confederations of diverse, loosely connected groups? What role did biological relatedness, being that of close kinship relations or long-term shared ancestry, play in the organization of these barbarian communities and how are such relationships related to patterns of material culture? Did this period involve long-distance migrations as described by late-antique authors?

One group with a relatively voluminous historical description are the Longobards (also known as Lombards, Longobardi, or Longbeards[9–11], see Supplementary Note 1 for more details). First described as living east of the lower Elbe River in the first century CE, Longobardi are reported around 500 CE north of the Danube, from where they then expanded into the Roman province of Pannonia (what is now western Hungary and lower Austria). In 568 CE the Longobard King Alboin led an ethnically mixed population into Italy, where they established a kingdom covering much of the country until 774 CE (Fig. 1a). One of the few contemporary texts describing the social structure and movement of the Longobards is by the Roman bishop Marius of Avenches who states "Alboin king of the Longobards, with his army, leaving and burning Pannonia, his country, along with their women and all of his people occupied Italy in fara"[12]. Etymologically the word *fara* is rooted in "to travel", but its precise meaning is ambiguous. While some have interpreted it to represent cognatic, kin-based clans, others suggest that it may simply refer to military units of mixed background[13].

Numerous sixth to seventh century archeological cemeteries in Pannonia and Italy contain broadly similar grave goods and burial customs, a pattern consistent with the historical account of a Longobard migration. In this study we generate paleogenomic data for 63 individuals from two of these cemeteries, Szólád in western Hungary and Collegno in northern Italy (Fig. 1a). We note that we are not aiming to infer Lombard ethnicity, which is a subjective identity. Our approach is unique in that we attempted to genomically characterize all of the interred individuals, rather than sampling individuals based on certain material culture markers. Combined with evidence of material culture, mortuary practices, and isotope data, our approach provides an unparalleled image of the social organization of these historical communities, and begins to shed new light on possible movements within Europe during this period.

## Results

**Two longobard-associated cemeteries.** We performed a deep genomic characterization of individuals buried in two cemeteries

of the sixth to seventh centuries CE that have material culture associated with the Longobards. Both are considered key sites with regard to the proposed migration from Pannonia to Italy. The first cemetery, Szólád, is located in present-day Hungary (Supplementary Figure 1). There are 45 graves (Fig. 1b), all of which are dated to the middle third of the sixth century based on a combination of stylistic elements of the grave goods and radiocarbon (2-sigma range of 412–604 CE, Supplementary Table 1) analysis[14]. Archeological, stable isotope, and mtDNA (HVS-1) analyses suggested that Szólád was occupied for only ~20/30 years by a mobile group of Longobard-era settlers[14]. The female to male ratio (sexing being based primarily on genetic data, or in its absence, anatomy) is 0.65. Graves in this cemetery are organized such that a core group ($N = 18$), mostly of male individuals, is surrounded by a half-ring of females ($N = 11$) (Supplementary Figure 2). Most of these individuals lie in elaborate graves with ledge walls and wooden chambers all in the same orientation, furnished with numerous artifacts such as beads, pottery, swords, and shields. The remaining 16 Longobard-period graves are more diverse in relation to the sex of the individuals, as well as to the quality of grave construction and richness of artifacts. Archeologists also recovered two bodies (AV1, AV2) that derive from a later occupation of the region by the Avars in the fill of the Longobard-period grave (SZ27)[15], as well the skeletal remains from an individual dating to the Bronze Age 10 m north of the cemetery (SZ1). See Supplementary Note 1, Supplementary Figures 1–11, and Supplementary Table 1 for the archeological context of Szólád.

The other cemetery, Collegno, is near Turin (Supplementary Figure 12), northern Italy, and was in use from the late sixth through the eighth centuries, the earliest period of the Longobard kingdom in Italy[16]. We studied the 57 graves that date between 580 and 630 CE based on artifact typologies (Fig. 1c) and that represent the first of three major periods of occupation. The types and range of grave goods in these 57 interments are comparable to those recovered at Szólád. However there is also evidence for a gradual cultural and religious evolution, with some practices disappearing in later decades. While there are no ledged graves, some are constructed via a wooden chamber structure, and there is the skeleton of a horse (devoid of head) in both cemeteries. See Supplementary Note 3 and Supplementary Figures 12–22 for the archeological context of Collegno.

Illumina sequencing of DNA extracts from petrous bone[17] and teeth identified 39 and 24 samples from Szólád and Collegno, respectively, for which there was high endogenous content, high library complexity, and patterns of postmortem damage (PMD) characteristic of ancient DNA (Supplementary Note 4, Supplementary Data 1). Analysis of X-chromosome-mapped reads in males and mtDNA in both sexes revealed low levels of estimated contamination in almost all samples (mean ~1%), although one, CL31, had a value of 27% and 7% using the X and mtDNA, respectively. While we include this sample in certain individual-based analyses, its results should be treated with caution. Genomic libraries for the majority of samples (60 out of 63) underwent partial UDG treatment[18]. Endogenous DNA content was sufficient (33–67%, mean 57%) for 10 male samples from Szólád to undergo whole-genome sequencing (WGS), with a mean genome-wide coverage across samples of 11.3×. The remaining 53 samples underwent an in-solution capture targeting 1.2 M single nucleotide polymorphisms (SNPs) (henceforth 1240 K capture)[19,20]. The average coverage at these SNPs (excluding the whole genomes) was ~1.5x and the mean number of genotyped SNPs per sample was ~522 K (Supplementary Note 5). Unless noted, we considered 33 and 22 samples from Szólád and Collegno for downstream analysis, respectively (three and one samples from Szólád and Collegno had fewer than 30,000 usable

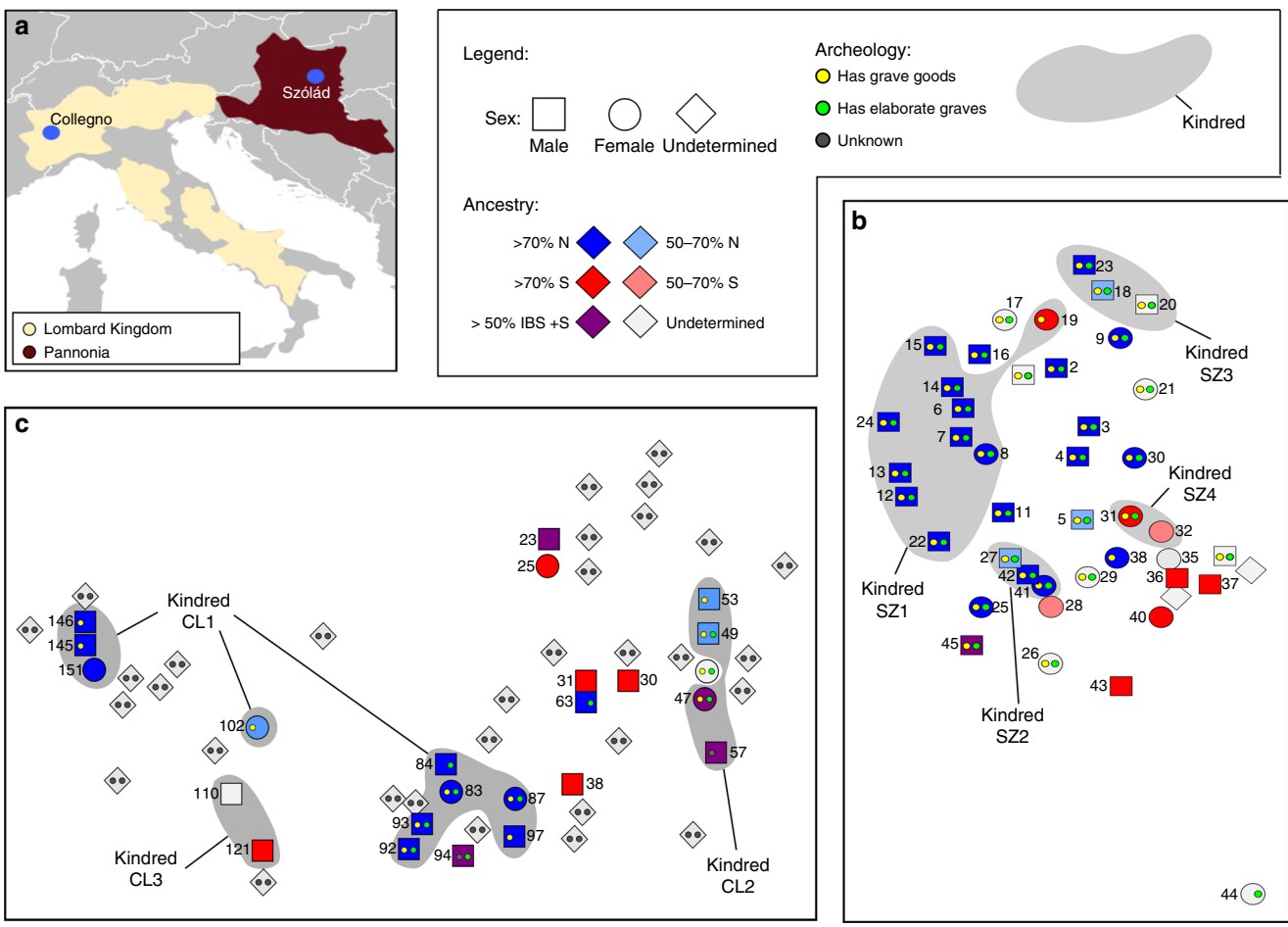

**Fig. 1** Archeological and genetic characterization of Szólád and Collegno. **a** Map of Europe showing the location (blue dots) of the two cemeteries and regional context is included (the Roman province of Pannonia in burgundy and the Longobard Kingdom in beige). **b**, **c** Spatial distribution of graves in Szólád and Collegno (first period burials only) with indication of sex (different shapes), genetic ancestry (different colors) and summary of archeology (yellow dots for presence/absence of grave furnishings and green dots for the presence of wooden elements in grave structure). Kindreds (in the biological sense) are indicated by gray shading in **b** and **c**. N = FIN + GBR + CEU, S = TSI. The map image in **a** is modified from the original Blank_map_of_Europe_ (with_disputed_regions).svg (https://commons.wikimedia.org/wiki/File:Blank_map_of_Europe_(with_disputed_regions).svg) by maix (https://commons.wikimedia.org/wiki/User:Maix) and is licensed under the Creative Commons Attribution-Share Alike 2.5 Generic license

SNPs, while samples SZ1, AV1, AV2, and CL36 were found not to belong to the same occupation period as the other samples).

In addition, we assembled comparative SNP data for different modern reference Eurasian samples, using directly genotyped SNPs from WGS[21,22] or by imputation[23,24], as well as 435 ancient West Eurasians from 6300 to 300 BC[20,25,26] (Supplementary Note 6 and Supplementary Data 2). We also included comparative genomic data from 18 Eurasian genomes[27–29] that are from similar time periods as the two cemeteries focused on here.

**Two primary central/northern and southern groups**. Principal component analysis (PCA) of samples from Szólád and Collegno against modern reference sets infers that our ancient samples possess genetic ancestry that overlaps overwhelmingly with modern Europeans (Fig. 2a, Supplementary Note 7, Supplementary Figures 23–25). However, they do not cluster with individuals from their respective modern countries of origin. Instead, samples from both cemeteries demonstrate a diverse distribution, with two broad clusters around modern northern and southern individuals, as well as individuals of intermediate ancestry. This north/south axis of genetic variation is also observed when

examining only our ancient samples, demonstrating that our results are not a bias introduced due to the reliance on modern reference populations or close kinship (Supplementary Figures 26–28).

We next analyzed our ancient samples using supervised model-based clustering analysis implemented in ADMIX-TURE[30] against a worldwide panel of modern reference samples (Supplementary Note 8, Supplementary Figures 29–43). The major genetic component in both Szólád and Collegno is CEU + GBR (it was difficult to consistently distinguish the ancestry coefficients for these two populations), with a mean of 64% and 57% across samples, respectively. TSI is the second most prominent component (mean of 25% and 33% across samples, respectively) (Fig. 3). An unsupervised analysis on a set of unrelated samples from Szólád and Collegno demonstrated a similar CEU + GBR vs. TSI- like structure (Supplementary Figures 44–46). By crudely assigning individuals to five color-coded groups based on relative ancestry components, a clear correspondence can be observed between our ADMIX-TURE and PCA analysis. Analysis of Y-chromosomes in males generally reveals a highly concordant pattern to the autosomes with regard to haplogroups that are most predominant in modern central/northern and southern Europeans

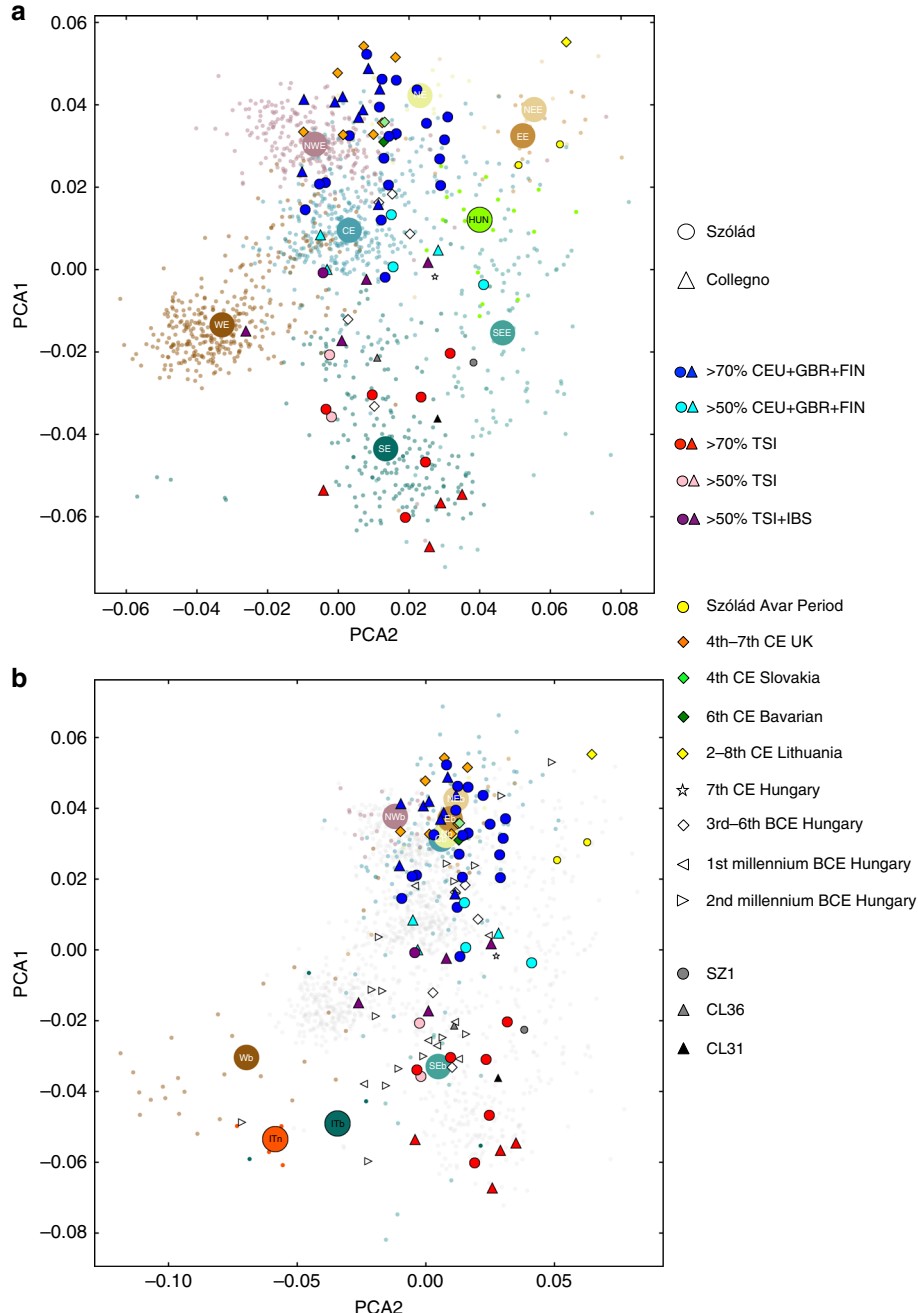

**Fig. 2** Principal component analysis of Szólád and Collegno. **a** Procrustes PCA of modern Europeans (faded small dots are individuals, larger circle is median of individuals) along with samples from Szólád (filled circles), Collegno (filled triangles) and other Migration Period samples. Szólád and Collegno samples are filled with colors based on estimated ancestry from ADMIXTURE. NWE, modern northwest Europe; NE, modern northern Europe; NEE, modern northeast Europe; CE, modern central Europe; EE, modern eastern Europe; WE, modern western Europe; SE, modern southern Europe; SEE, modern southeast Europe; HUN, modern Hungarian. **b** Procrustes PCA of modern and Bronze Age Europeans along with samples from Szólád and Collegno and other Migration Period samples. Gray dots are modern Europeans as shown in **a**. NWb, Bronze Age northwest Europe; Nb, Bronze Age northern Europe; NEb, Bronze Age northeast Europe; Cb, Bronze Age Europe; Eb, Bronze Age Europe; Wb, Bronze Age western Europe; SEb, Bronze Age southeast Europe; ITb, Bronze Age Italy; ITn, Neolithic Italy

(Supplementary Note 9, Supplementary Figures 47–49, Supplementary Data 3, Supplementary Table 2).

A population assignment analysis (PAA) that estimates uncertainty in genetic ancestry assignment finds that individuals with high CEU + GBR ancestry are assigned to countries from all over modern central, northern, and northwest Europe (Supplementary Note 10, Supplementary Figure 50, Supplementary Data 4). We refer to this as central/northern ancestry as it is

generally difficult to distinguish this with more precision given the resolution of our data. A series of D-statistics analyses of the form D (ancient, ancient, modern_reference, outgroup) confirms the close relationship amongst our ancient samples assigned as being of primarily central/northern ancestry from both Szólád and Collegno (there are even four Szólád-Collegno pairs that appear to form significant clade compared to all other modern populations), while those of southern ancestry show more

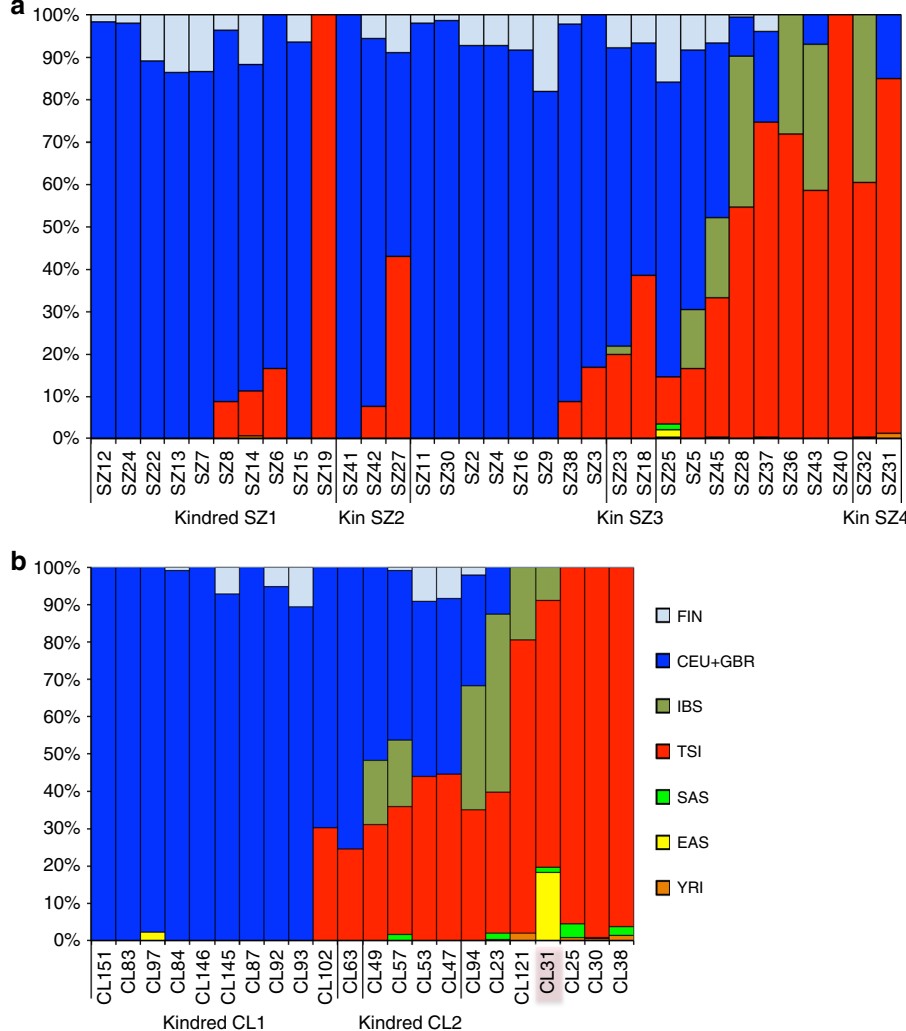

**Fig. 3** Genetic structure of Szólád and Collegno. Model-based ancestry estimates from ADMIXTURE for Szólád (**a**) and Collegno (**b**) using 1000 Genomes Project Eurasian and YRI populations to supervise analysis. Note that high contamination was identified in CL31 and overlaid with a pink hue in **b**

potential structure (Supplementary Note 11, Supplementary Figures 51–60, Supplementary Table 3). An analysis of rare variants[27] in our nine medieval whole genomes from Szólád is consistent in terms of relative amounts of central/northern and southern ancestry, demonstrating that our results are not the result of any SNP ascertainment bias (Supplementary Note 12, Supplementary Figures 61–64, Supplementary Table 4).

We also examined our ancient samples within the context of the prehistoric groups that were the major contributors to modern European genetic variation: Paleolithic hunter-gatherers (WHG), Neolithic farmers (EEF), and Bronze Age Steppe herders (SA). Both PCA and supervised and unsupervised ADMIXTURE analyses (Supplementary Figures 65-72) essentially reiterate the same structure amongst our ancient medieval samples, with greatest EEF ancestry in those individuals demonstrating similarities to modern southern Europeans and greater WHG + SA ancestry in those that resembled modern northern Europeans (with WHG being predominant in northwest Europe and SA in northeast Europe).

Relating our results to questions of migration requires us to understand to what extent the geospatial distribution of genetic diversity today reflects that of the Migration Period ~50–60 generations ago. While previous sampling from the era has been limited, we note that published fourth- to seventh-century

genomes from Britain, Bavaria, Lithuania, and the Caucasus, analyzed alongside our own ancient samples, cluster close to their modern counterparts. The next temporally closest major European sample to the Migration Period involves a large number of recently characterized Bronze Age individuals that are ~100 generations separated from the Migration Period[20,25,26]. Though there are discrepancies, we find a general genetic similarity between individuals sampled from the same location today and in the Bronze Age at a continent-wide scale when considering northern and southern ancestry (Fig. 2, Supplementary Figures 73, 74), suggesting that the strong isolation-by-distance pattern observed in modern day Europeans was emerging ~4000 years ago (and presumably would have been even more similar ~1500 years ago). Based on PCA and D-statistic analyses (Supplementary Figures 75–84), individuals from Szólád and Collegno with high CEU + GBR ancestry that make up the majority of our sample are significantly closer to Bronze Age central, northwestern, eastern (Polish), and (at least using PCA) northern Europeans than Bronze Age Hungarians. We found no evidence that such ancestry was present in northern Italy during this time (who instead resemble modern southern and Iberian Europeans), which would be consistent with inferred long term barriers to gene flow in Europe across the Alps[31]. As noted previously[26], Bronze Age populations and Hungarian

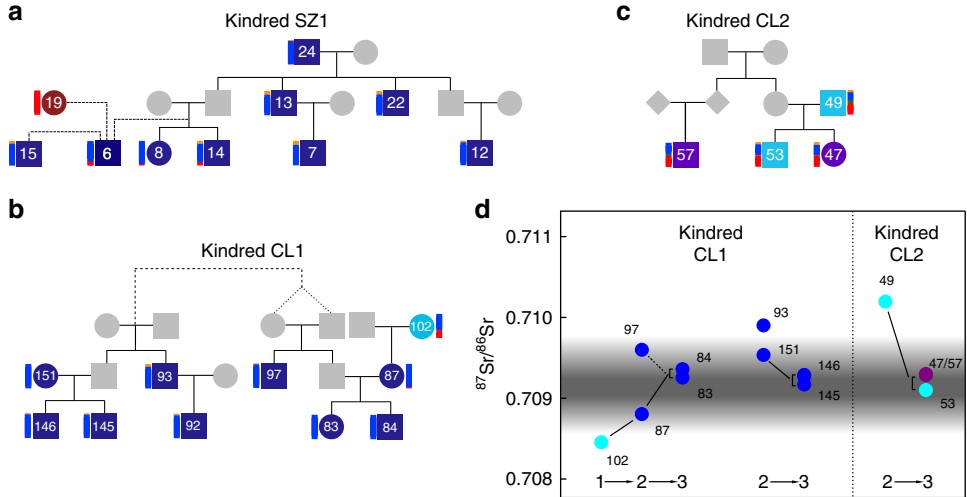

**Fig. 4** Major kindreds in Szólád and Collegno. Kindreds SZ1 (**a**), CL1 (**b**), and CL2 (**c**), with colors corresponding to criteria and labeling in Fig. 2. ADMIXTURE coefficients (vertical bars on side of each individual) estimated using 1000 Genomes[21] European populations only. Dashed lines indicate relationships of unknown degree (including past inbreeding in the parents of CL97). **d** Displays strontium isotope ratios for Kindreds CL1 and CL2 where available. Estimate of local range at Collegno is shown using black shading, as detailed in Fig. 4. Numbers indicate generations (1 being the oldest, 3 being the youngest)

Scythians from the third to sixth centuries BCE are diverse, with most sharing similarity with modern southern Europeans, though a minority are found in close proximity to the central/northern samples from Szólád and Collegno in the PCA. Overall, we suggest that based on modern and Bronze Age data, the high CEU + GBR ancestry observed in both Szólád and in particular Collegno is unusual.

**Both cemeteries are organized around biological kinship**. We utilized lcMLkin[32] to infer pairwise biological relatedness within ancient cemeteries at an unprecedented level compared to previous studies[33,34] (Supplementary Note 13, Supplementary Figures 85–87, Supplementary Data 5). We hereafter refer to groups of biologically related individuals as kindred as a shorthand, though we recognize that kinship in the traditional archeological/anthropological sense encompasses a much broader range of social relationships[35].

Within Szólád we identified four kindreds among the Longobard era burials (gray shadings in Fig. 1b), with one particularly large one (Kindred SZ1, Fig. 1b). Kindred SZ1 (Fig. 4a) spans three generations and consists of ten individuals in close spatial proximity. Seven individuals all share recent identity-by-descent (IBD) from SZ24 (one of the oldest individuals in the cemetery, between ~45 and 65 years old (yo.)), while another two individuals, SZ15 and SZ19, are more remotely connected genealogically to the kindred via SZ6, a young male aged 8–12 yo. at the time of death. While SZ6 is related to all other individuals in this kindred, sharing the greatest IBD with siblings SZ8 and SZ14, we are unable to determine the exact genealogical relationships involved, likely because of its low SNP coverage (0.048×).

Individuals in this kindred were buried with a rich diversity of grave goods, and all but one was buried in elaborate ledge graves. Only two members of this kindred are female, SZ8 and SZ19, who are estimated to be 3–5 and 17–25 yo. respectively at death. The rest are males, aged from ~1 to ~65 yo. These graves occupy a prominent position in the northwest of the cemetery, with all but SZ19 found amongst the core group (she is instead found in the external half-ring of women; Supplementary Figure 2). Six male individuals in this kindred were buried with weapons, despite

three (SZ7, SZ14, and SZ15) being teenagers at the time of death (12–17 yo.). The adult males in this kindred (SZ24, SZ13, and SZ22) appear to have had access to a diet particularly high in animal protein, as inferred from nitrogen isotope analysis[14]. Individual SZ13 has the deepest grave and is the only individual in the whole cemetery whose burial includes a weighing scale and a horse, which may be an indication of his differentiated status in that society. We note that this kindred lacks adult female descendants of SZ24, though we were unable to sample some of the female graves in the half-ring structure (unsampled graves SZ21 and SZ29 could still be potential mothers of the third generation; individuals SZ17 and SZ26 can be excluded based on mtDNA analysis by Vai et al.[36]).

While individuals in this kindred are predominantly of a central/northern European genetic ancestry, they are not genetically homogenous. Again, SZ19 is an outlier, strikingly possessing 100% TSI ancestry. In addition SZ6 and the two third-generation siblings SZ8 and SZ14 also possess a small but noticeable TSI ancestry component. Assuming that one of the siblings' parents represented the central European ancestry seen in their two uncles (SZ13 and SZ22), we inferred (using an adapted version of spatial ancestry analysis (SPA) Supplementary Note 14, Supplementary Figures 88–96) that the other parent likely possessed an ancestry that most resembled modern day French individuals (Supplementary Figures 92–93). This latter individual would probably have been female, as while SZ14 has a similar Y chromosome to SZ13 and SZ22, both siblings possess a different mtDNA type to their uncle's (I3 vs. N1b2).

In Collegno, we identified three kindreds, with one particularly extensive one. Nine of the ten individuals from the largest kindred (Kindred CL1; Figs. 1c and 4b) were buried in elaborate graves and/or with artifacts. In contrast to Szólád, individuals with close biological relationships occupied spatially distant graves. Interestingly, the spatial cluster with six individuals is chronologically older (570/590-610 CE) than the more westerly trio (Supplementary Figure 19). Kindred CL2 (Fig. 4c) is found in the east part of Collegno, with graves positioned in a row running north to south (Fig. 1c).

Similar to Kindred SZ1, Kindred CL1 is predominantly of central/northern European ancestry. However, while genetically quite similar, on average members of this kindred possess slightly

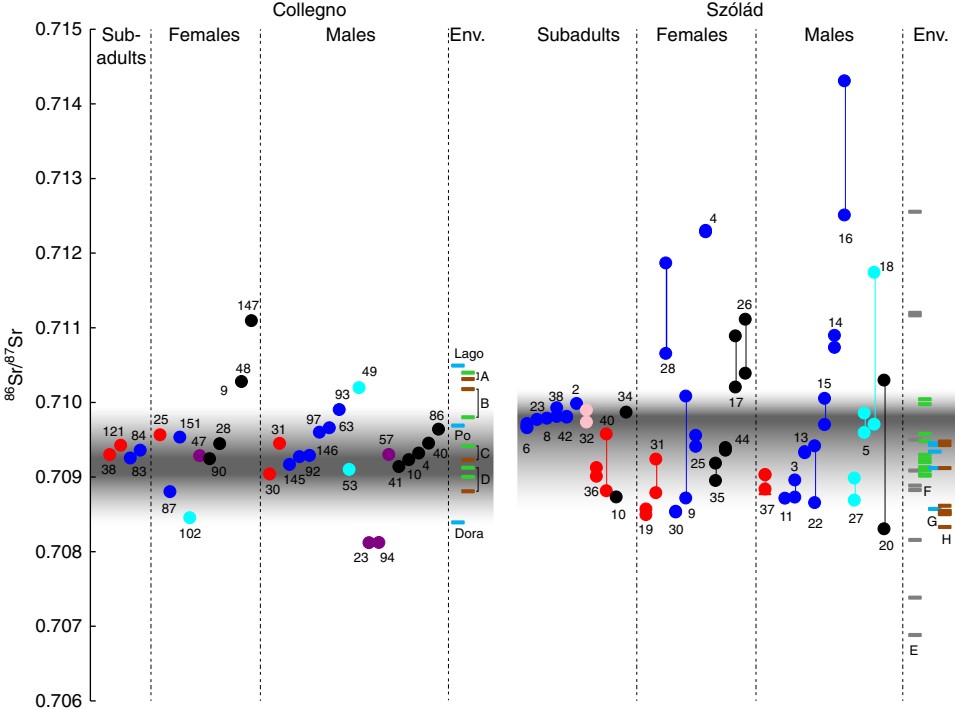

**Fig. 5** Strontium isotope ratios in Szólád and Collegno. Evidence for migrants at Collegno and Szólád, as suggested by $^{87}$Sr/$^{86}$Sr values of human tooth enamel and environmental reference samples. Isotopic data from Szólád were taken from the ref.[14]. The vertical lines indicate where multiple teeth were sampled from the same individual. The gray band denotes the local bioavailable strontium isotope range. Individuals falling outside of this are considered non-local. A: Collino di Superga (clays and marls); B: Lago Piccolo di Avigliana (glacial deposits); C: Castello di Avigliana (serpentinite, green schist); D: Collegno site (Pleistocene gravels); E: Bakony Mountains; F: vegetation south of Lake Balaton; G: water south of Lake Balaton; H: soil near Szólád (Szólád data from the ref.[14]). Color coding scheme corresponds to ancestry criteria and labeling in Fig. 2, though samples represented in black are those lacking genomic data. To ease readability of the figure, the $^{87}$Sr/$^{86}$Sr values of each ancestry group are arranged in ascending order

less FIN ancestry and are thus more shifted towards northwestern Europe in the PCA, SPA, and PAA. In addition, this group is again not genetically homogenous, with the unsampled father of CL87 being of much greater central/northern European ancestry than the mother, CL102, who has an ancestry profile again most consistent with modern day France (Supplementary Figure 95). Kindred CL2 also has a more mixed genetic ancestry based on the ADMIXTURE analysis that would generally be associated with a more modern central European ancestry than Kindreds SZ1 and CL1. Interestingly the grave goods of the daughter, CL47, and an unsampled adjacent female, CL48, resemble burials from this time in southern France and Switzerland (Supplementary Note 3). Members of these two large kindred groups also appear to have generally consumed more animal protein than other individuals in the cemetery (Supplementary Figure 97).

**Ancestry is associated with elements of material culture**. In both cemeteries individuals with predominantly central/northern and southern European ancestry possess very distinctive grave furnishings. In order to quantify this relationship, we classified individuals into either Northern (N) or Southern (S) groups based on their proportion of CEU + GBR + FIN ancestry versus TSI + IBS ancestry, and used these to conduct a series of Fisher's exact tests for their association with material culture (we note that our results were robust to our specific ancestry cutoffs; see Supplementary Note 15, Supplementary Table 5). We focused on artifacts potentially associated with either specific cultural traditions (e.g., S-brooches and stamped pottery) or individual profession or status (e.g., war weapons). In both Szólád and Collegno individuals with N ancestry were significantly more often buried with

grave goods ($p$-value < 0.0071, Fisher exact test, Supplementary Table 6). In contrast, no S individual was buried with such artifacts, with only two exceptions (females SZ19 and SZ31). This association between genetic ancestry and material culture is particularly significant for beads (from necklaces and pendants) and food offerings in Szólád, as well as weapons in both Collegno and Szólád (Supplementary Table 7). We note that one artifact in grave SZ19 is stylistically distinct (possibly Roman) from the artifacts found in other graves in the same cemetery. Grave type ($p$-value < 0.02, Fisher exact test, Supplementary Table 8) also significantly differs between groups in both cemeteries, with N individuals presenting graves with wooden elements, as opposed to simple pits (more common amongst graves with S individuals).

**Comparison of genetic and strontium isotope data**. We also generated new strontium isotope data ($^{87}$Sr/$^{86}$Sr) for Collegno to complement the existing data from Szólád[14] and analyzed them within the context of our genomic data in order to better understand patterns of immigration to these two sites (Supplementary Note 16, Supplementary Figures 97–104, Supplementary Tables 9-13). Within Szólád we find that adult individuals with both predominantly central/northern and southern genomic ancestry possess similar non-local signatures (Alt et al.[14] described this as Range I) (Fig. 5). This might suggest that individuals from both ancestry groups immigrated into Szólád together despite the differences in material culture. However, we also note generally a quite diverse non-local range amongst adults with central/northern ancestry (for example SZ4 and SZ16 are extreme outliers), suggesting that not all individuals originated from the same location prior to settling in Szólád.

In contrast, in Collegno it was notable that the five individuals with major southern ancestry primarily assigned to Italy using PAA exhibited local strontium isotope signatures. When examining the two major kindred, we observe the striking general pattern that earlier generations had strontium isotope values that diverged from the local range more than later generations (Fig. 4d, Supplementary Figure 100). This appears to fit a model of individuals of central/northern European ancestry migrating and settling in Collegno amongst a set of local individuals of primarily Italian origin.

## Discussion

The most striking feature of our data is the inference of two main clusters of genetic ancestry that are shared amongst our two sixth- to seventh-century cemeteries separated by almost 1000 km. In both Szólád and Collegno this genetic structure mirrors the variation that emerges from their mortuary practices, i.e., how living members of the community represented the individuals that they buried. This perhaps suggests that in these two cemeteries there may indeed have been a biological basis to the notion that long-term shared common descent can shape social identity and that this is reflected in the material culture. However, whether the association between genetic ancestry and material culture reflects specific peoples mentioned in historical texts (i.e., Longobards) or stemmed from a deeper/long-term descent (of mixed barbarian ancestries) is as yet unclear. Future research may reveal how widespread such a link between ancestry and material culture was in the Migration Period.

Our genomic characterization of Szólád and Collegno provides novel insight into the structures and hierarchies of societies from the Migration Period in two very different contexts. On the one hand, a previous study of Szólád had identified this cemetery as belonging to a highly mobile community that was settled in the region for approximately one generation. We were able to show in particular that the burials in this cemetery were organized around a three-generation kindred with ten members (Kindred SZ1), all but one of which were of predominantly central/northern ancestry. Members of this kindred stand out in relation to others for a number of reasons: (i) access to diet higher in animal protein; (ii) graves occupying a prominent position in the cemetery; (iii) the presence of the oldest individual in the cemetery (SZ24); and (iv) the individual (SZ13) with the deepest grave and the only one buried with a horse, Thuringian type pottery, and a scale with weights and coins.

Surrounding this kindred within the area demarcated by the half-ring of women there are ten males. Of the nine for which we have genetic data, all have predominantly central/northern European genetic ancestry but some have both additional TSI and IBS ancestry (for example SZ5), suggesting a somewhat diverse genetic makeup and thus possibly different geographic origins from each other and compared to the focal Kindred SZ1. All adults and teenagers have weapons, and three out of four adults share the same non-local Sr signature as adults in Kindred SZ1. The half-ring of women itself is made up of a mixture of individuals (ten adults and one child), with five that have predominantly central/northern ancestry and three that have majority TSI + IBS ancestry. They also have a wide range of strontium isotope ratios. The female SZ19 in this half-ring is part of Kindred SZ1. She lacks the ledge graves of the other members, has a distinct material culture, and is of southern ancestry. To what extent her lack of IBD with SZ24 and her different ancestry contributed to her more peripheral burial compared to both the female SZ8 (who is buried within the core) and the male SZ15 (who is similarly unrelated to SZ24 but again found within the core) is an open question.

Since the adults were almost all non-local, it is tempting to suggest that we may be observing the historically described fara during migration. Regardless, this group appears to be a unit organized around one high-status, kin-based group of predominantly males, but also incorporating other males that may have some common central/northern European descent. The relative lack of adult female representatives from Kindred SZ1, the diverse genetic and isotope signatures of the sampled women around the males and their rich graves goods suggest that they may have joined the unit during the process of migration (perhaps hinting at a patrilocal societal structure that has been shown to be prominent in Europe during earlier periods[37]).

The remaining part of this community for which we have genomic data ($N = 7$) is composed of individuals of mainly southern European genetic ancestry that are conspicuously lacking grave goods and occupy the southeastern part of the cemetery; they were buried in randomly oriented graves with straight walls. While the lack of grave goods does not necessarily imply that these individuals were of lower status, it does point to them belonging to a different social group. The strontium isotope data suggest that they may have migrated together with the warrior-based group from outside Szólád, but barriers to gene flow were largely maintained.

In contrast to Szólád, Collegno likely reflects a community that settled for multiple generations. Here, organization around at least one large extended immigrant kindred once again seems to have been a key element of social organization. However, there is more spatial variation, with the kindred spreading outwards from the center point of the cemetery over time. There is also one other significant immigrant kindred of a different genetic origin and material culture that also holds a central position in the cemetery, and, for this first period of occupation at least, these two groups appear to remain distinct genetically. Unlike at Szólád, we find evidence of only one other individual (CL63) of predominant central/northern ancestry who does not belong to the major kindred unit (though our sampling of this cemetery is not as complete as Szólád). On the other hand, individuals of southern ancestry of the type that would typically be found in the region today appear to be local to the Collegno area based on isotope data, show much more scattered burials and are poorer with regard to grave goods and animal protein consumption. As such it is tempting to infer a scenario of these large immigrant barbarian families exerting a dominating influence on the original resident population.

Our two cemeteries overlap chronologically with the historically documented migration of Longobards from Pannonia to Italy at the end of the sixth century. We observe that central/northern European ancestry is dominant in both Szólád and Collegno, and that modern genetic data do not show a preponderance of such ancestry in either Hungary or northern Italy. While we do not yet know the general genomic background in these geographic regions before the establishment of Szólád and Collegno, other Migration Period genomes show a fairly strong correlation with modern geography. Going further back in time, samples from Szólád and Collegno with high central/northern ancestry are genetically closer to Bronze Age populations north of Hungary than of Hungary itself. Hungary has demonstrated a diverse and shifting genetic profile from the Bronze Age through to today. Observing some mix of resident individuals with both primarily central/northern and southern ancestry in Szólád seems plausible. However, the observation of a majority of individuals with central/northern ancestry in modern or Bronze Age northern Italy is unexpected. Analyzing the paleogenomic data alongside the strontium isotope data further favors a migration hypothesis, as this indicates that the earliest individuals of central/northern ancestry in Collegno were probably migrants while

those with southern ancestry were local residents. Our results are thus consistent with an origin of this group east of the Rhine and north of the Danube and we cannot reject the migration, its route, and settlement of the Longobards described in historical texts. This is also consistent with a parallel study that explicitly modeled this scenario using whole mitochondrial genomes[36].

Modern European genetic variation is generally highly structured by geography[23,38,39]. It is surprising to find significant diversity within small, individual cemeteries. Even amongst the two family groups of primarily central/northern ancestry there is clear evidence of admixture with individuals with more southern ancestry. Whether these people identified as Longobard or any other particular barbarian people is therefore impossible to assess. If we are seeing evidence of movements of barbarians, there is no evidence that these were genetically homogenous groups of people.

We have observed an intriguing association between genetic evidence, isotope data and material culture that sheds new light on the social organization of sixth-century communities during both migration and settlement phases. A key aspect of our approach is the in-depth sampling of entire cemeteries. We propose that this is a conceptual and methodological advance: in the future one must use a similar whole cemetery-based genomic methodology to explore whether the results observed here are common to other sites from late antiquity and the early middle ages. The genetic complexity observed within these cemeteries presents a new set of questions concerning population structure within past societies. Moreover, if the genetic similarity we observe between Szólád and Collegno appears in other contemporary cemeteries, we will be able to better appreciate the extent and dynamics of these movements and of the invasion of barbarian peoples across the Roman Empire.

## Methods

**DNA sequencing and bioinformatic processing**. Bone specimens from Szólád and Collegno were prepared in clean-room facilities dedicated to ancient DNA in the Laboratory of Molecular Anthropology and Paleogenetics, University of Florence. DNA extraction was performed using a silica-based protocol[40]. Genomic libraries were prepared at the University of Florence and at the Max Planck Institute for the Science of Human History in Jena according to modified Illumina protocols and were screened for endogenous human DNA via mapping to the human reference genome after re-sequencing. DNA postmortem damage (PMD) patterns typical of ancient DNA were assessed with MapDamage[41]. The ten samples (all from Szólád) with best DNA quality and concentration were submitted for WGS on an Illumina HiSeq 2500 1TB at the New York Genome Center. Another 53 samples underwent a capture and NextSeq sequencing for ~1.2 M SNPs for an average coverage of ~1x per sample. Reads were trimmed, merged (where applicable), mapped, and filtered for PCR duplications according to a protocol optimized for ancient DNA[42,43]. Genotype likelihood estimation and pseudo-haploid and diploid calling were performed using in-house Python scripts (www.github.com/kveeramah). For samples subject to partial UDG treatment, genotype likelihood estimation was implemented, ignoring the first and last three bases of reads. Genotype calling considered PMD for the remaining samples[44]. Finally, an additional pseudo-haploid call set was generated by sampling a random read at each position. See Supplementary Notes 4 and 5 and Supplementary Table 14 for further details.

**Modern and ancient reference samples**. We assembled SNP data matching the 1240 K capture for three modern reference datasets and one ancient reference dataset for comparison to the early medieval samples generated in this study. The three modern reference sets consisted of an imputed[45] European POPRES[46] SNP set with ~300 K SNPs, an imputed Eurasian[24] SNP set with ~700 K, and a 1000 Genomes[21] and SGDP[22] whole genome set matching all 1240 K SNPs. Our ancient reference set consisted of 435 ancient West Eurasians from 6500 to 400 BCE[20,25,26] with calls made at the 1240 K capture SNPs. Depending on the context, we made pseudo-haploid calls by randomly drawing one allele from a diploid genotype.

**Biological relatedness inference**. We used the software lcMLkin[32] to estimate biological relatedness and coancestry coefficients for every pair of samples. In addition to using the allele frequencies of the ancient samples themselves, we also

adapted the software to utilize allele frequencies from other sources, in this case the CEU and TSI 1000 Genomes populations, and incorporate admixture.

**Principal component analysis**. PCA of SNP data was conducted using smartpca[47]. When analyzing Migration Period individuals against reference populations, individual pseudo-haploid PCAs were conducted for each ancient sample separately, and individual analyses were then combined using a Procrustes transformation in R using the vegan package[48]. LD pruning was performed using the indep-pairwise function in PLINK[49].

**Model-based clustering analysis**. Supervised and unsupervised model-based clustering was performed using ADMIXTURE[30]. Dependent on the analysis, target Migration Period samples were analyzed individually (to avoid the effects of relatedness) or together (using a set of unrelated individuals that maximized SNP number).

**Population assignment analysis**. PAA was conducted as described in Veeramah et al.[28] using the POPRES and HellBus datasets.

**D-statistic analysis**. D-statistic analysis as described in Patterson et al.[50] was performed using custom Python software that allowed multithreading.

**Spatial ancestry analysis**. We applied the software SPA[51] to analyze the POPRES imputed SNP dataset. We also further extended the software to allow the use of pseudo-haploid calls in our ancient samples, and to infer the location of one parent of an admixed individual given the known location of another parent.

**Rare variant analysis**. We followed the approach of Schiffels et al.[27] to examine the relative sharing of central/northern European and southern European-specific rare variants, and used the software rarecoal to assign our ancient whole genomes to a branch on a bifurcating demographic model based on the analysis of rare variants in modern European populations (TSI, IBS, and GBR from the 100 Genomes project[21], Denmark[52], and the Netherlands[53]). Analysis was performed using both pseudo-haploid calls and diploid genotypes.

**Y chromosome analysis**. The phylogenetic position of each Y chromosome variant observed in the whole sample was established according to its occurrence in public database or in the published literature[54,55]. The lack of base calls due to the absence of reads at a position in a particular sample was resolved either as an ancestral or derived allele by a hierarchical inferential method according to the phylogenetic context based on a cladistics approach. The phylogenetically informative SNPs were used to build a parsimony-based phylogenetic tree using the Pars application of the Phylip v3.69 package. FigTree v1.4.2 software was used to display the generated tree.

**Isotope analysis**. Strontium, oxygen, carbon, and nitrogen isotope analysis was carried out on 33 first-period individuals from Collegno, together with faunal and environmental reference samples. Tooth enamel powder primarily from second premolars or second molars was used for strontium isotope analysis at the Isotope Geochemistry Laboratory of the Department of Earth Sciences, University of Cambridge. Preparation of tooth enamel powder for oxygen isotope analysis followed the method described by Balasse et al.[56]. Collagen was extracted from bones for nitrogen and carbon isotope analysis, based on the method detailed by Privat et al.[57]. These analyses were carried out at the Godwin Laboratory, University of Cambridge. To determine local bioavailable strontium values, samples of water, vegetation, and soil were collected both at the site of the cemetery of Collegno and at locations considered to be about a day's walk from the site and then pre-treated following the procedures outlined in Maurer et al.[58]. Faunal samples were taken from six species from a third-/fourth-century site in Piazza Castello in Turin to provide an ecological baseline for human diet.

**Code availability**. Code generated to call variants in the ancient samples is available at https://github.com/kveeramah/.

## Data availability

Sequencing data (as processed BAM files) are available from the NCBI sequence read archive (SRA) database under accession # SRP132561 (1240 capture data) and SRP132581 (WGS data). The collections and methods for the Population Reference Sample (POPRES) are described by Nelson et al.[46]. The datasets used for the analyses described in this manuscript were obtained from dbGaP at https://www.ncbi.nlm.nih.gov/projects/gap/cgi-bin/study.cgi?study_id=phs000145.v4.p2 through dbGaP accession number phs000145.v4.p2. All other relevant data are available upon request.

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

## Acknowledgements

We thank Kurt Alt for his role in scientifically characterizing Szólád. We are particularly appreciative of discussions with Kathryn Twiss, Philipp von Rummel, Falko Daim, Frans Theuws, and Helmut Reimitz. We are grateful to Hazel Chapman and James Rolfe for help with isotopic analyses and to David Redhouse for help with an illustration. We thank Marta Burri and lab members of the Max Planck Institute for the Science of Human History in Jena for laboratory support. We thank Ágnes Kustár and Ildikó Pap at

the Department of Anthropology of the Hungarian Natural History Museum, Budapest, for providing the anthropological material from the cemetery of Szólád. This work was supported by National Science Foundation award #1450606, the Anneliese Maier Research Award of the Alexander von Humboldt Foundation, the Max Planck Society, the German Federal Ministry for Education and Research, the Swedish Riksbankens Jubieleumfond, the Gerard B. Lambert Foundation, the Institute for Advanced Study Director's Office, and the Italian Ministry for University and Research Department of Excellence Program.

## Author contributions

I.K., L.P.B., E.B., C.G., T.V., D.W. and U.F. provided the archeological material and/or performed the archeological analysis and interpretation. M.C.L.R., W.P. and P.J.G. provided the historical background and interpretation. S.V., C.P., A.M., B.M., M.L., J.K. and D.C. performed the ancient DNA lab work and screening. C.E.G.A., D.B., C.P., S.G., G.B. and K.R.V. and performed the downstream bioinformatics and population genetic analysis. P.F. performed the Y-chromosome analysis. S.H. performed the isotope analysis. C.E.G.A., P.J.G. and K.R.V. wrote the paper. W.P., P.J.G. and K.R.V. conceived the study.

## Additional information

**Competing interests:** The authors declare no competing interests.

