## [Peer Review File · Nature Communications]

Reviewer #1 (Remarks to the Author):

In Amorim et al., they study the genomic characteristics of individuals in two medieval cemeteries from w. Hungary (Szolad) and n. Italy (Collegno), archaeologically associated with the Longobard barbarian tribe. They wanted to test for the existence of a long distance migration into northern Italy by these people and to examine the genetic structure of these cemetery communities. They studied the genetic ancestry of as many interred individuals as possible and further studied the material culture, mortuary practices and isotope data associated with the remains. At both cemeteries, there were individuals of more 'northern European' (NE) ancestry and individuals of more 'southern European' (SE) ancestry, as well as identifiable family groups, with one predominant family group in each. The ones belonging to the large family group tended to be associated with NE ancestry. Using strontium isotope analysis, they showed that at Szolad, adults associated with both ancestries show diverse non-local ranges, while at Collegno older generations within the kin group had non-local strontium signatures, while more recent generations and those associated with southern European ancestry exhibit local strontium signatures.

The results of this paper are very interesting, especially the pedigree analysis and correlations in genetic and material/isotopic data, but it could benefit both from some reorganizing, additional genetic analyses, and a better discussion of how to interpret the analyses regarding genetic ancestry. Some major comments:

(1) Restructure introduction to address questions you actually answer. Current proposed questions (p. 3, bottom) are not answerable as they address all barbarian tribes, not just the Longobard, and even in the Longobard paragraph (p. 4), focus is on the long distance migration, which is not sufficiently discussed (perhaps because not enough evidence?). Move more quickly to sampled cemeteries and specific questions addressing them – i.e., genetically who they are, internal genetic patterns to arrive at an understanding of kinship patterns, connection between genetics and grave good/elaborateness distribution, isotopic analysis and correlation to genetics/family. Perhaps a longer discussion on implications for understanding barbarian tribes can be included at the end, but that is not what this study addresses.

(2) The bulk of the genetic section (p. 6-7) reiterates the same conclusion of northern-southern ancestry and takes up a large part of the paper. However, the discussion doesn't then dive into an exploration of whether long distance migration is the best explanation, and in fact back pedals with the sentence "However, whether the association between genetic ancestry and material culture reflects specific peoples mentioned in historical texts (i.e. Longobards) or stemmed from a deeper/long-term descent (of mixed barbarian ancestries) is as yet unclear (p. 11, bottom)". The rest of the discussion, however, seems to imply long distance migration was likely, particularly in Collegno (p. 13). And, what does it mean that in Szolad, we also see individuals of predominantly SE ancestry? You suggest they brought many foreign individuals to that area, but does that connect to

any historic text and does it weaken conclusions for Collegno? I think this section is better served by having a shorter summary of the consistent genetic results and then detailing the different scenarios that could give rise to such genetic patterns (and what scenarios cannot). Then, it seems the non-genetic analyses, particularly the isotope one, can be used to help favor one hypothesis over another.

(3) A large assumption here is that present-day distributions do reflect what the distribution should look like during this time period. I am less convinced without ancient data to help confirm this. For instance, based on the genetic analysis, I could see an argument that northern Italy was predominantly of NE ancestry at the time, and more and more people of SE ancestry entered that area, leading to much more SE ancestry today. I don't necessarily believe that is true, but I don't think that demographic model has been eliminated. A deeper look into the validity of this assumption is worth having, and if possible, using more published ancient data to determine whether the assumption is valid would be useful.

(4) For the PCA analysis, the supplemental text says: "It is also noteworthy that samples from Szólád do not appear to be clustered next to Neolithic, Bronze Age or modern samples from Hungary. This lack of overlap with the Bronze Age is made more apparent by performing this analysis only with Bronze age, Medieval and modern samples" (p. 56). However, at least in Figure S7.4, if I interpret it correctly, the BB (Bell Beaker), EBr (Eastern European Bronze Age), HBr (Hungarian Bronze Age), NBr (northern European Bronze Age) at the least all seem to overlap with both some Collegno and Szolad samples. Thus, there are overlaps with Bronze Age samples, and I think this helps your argument of more central/northern European ancestry in some samples (from both cemeteries). I think this is important to describe more clearly, both in the supplement and main text.

(5) In the Mathieson et al. 2015 paper, there are 2-3 samples in their Figure 1a that are found in northern Italy. I'm not sure who those individuals are, but I think it is worth highlighting them, as if they have northern/central European ancestry and are older, then it would suggest there was already such ancestry in this region previously, making it more difficult to argue that the presence of northern ancestry is related to a Lombard migration.

(6) Most Admixture analyses are unsupervised, including samples from potential reference populations and projecting samples with few data (like most ancient individuals). Why was an unsupervised analysis, estimating the CVV to determine the best K groups, not done? I would want to see an unsupervised analysis to make sure unknown biases are not being introduced with the supervised analysis.

(7) D-statistics: The major analyses assigning genetic ancestry are all based on more exploratory analyses. I believe there is enough data here that you can do D-statistics (and f3-statistics) to more directly test which connections these ancient individuals share to each other and other European populations. Testing $D(\text{ancient}, X; \text{ancient}, \text{Mbuti/Chimp})$ would help establish whether these ancient individuals form a clade with each other relative to other populations/individuals (X), and what types of admixture is characteristic of this population. Adding $D(\text{ancient}, \text{ancient}; X, \text{Mbuti/Chimp})$ where X are southern Italians/Spanish or northern Hungarians/central Europe populations/individuals would be the best confirmation of conclusions of asymmetric NE/SE ancestry made in ADMIXTURE/PCA.

Minor comments:

- (1) In the main text, especially the abstract, make sure it is clear what terms such as barbarian, medieval, Pannonia, refer to for the average reader.
- (2) On p. 5, please list mean and range of radiocarbon dates directly in main text.
- (3) The PCA figures are very hard to read. I would suggest making modern individuals gray-scaled, and using triangles instead of stars to reduce the number of edges. I would also consider not using the border color/thickness – it's a subtle detail that is very hard to distinguish.
- (4) Are the results for the Bronze Age samples relative to present-day populations consistent with previous analyses? Is there any ascertainment bias in using the Mathieson et al. data?
- (5) Published ancient data cites Mathieson et al. (2015), but I think most of their data came from a previously published dataset in Lazaridis et al. 2014 and Allentoft et al. 2015.
- (6) Any thoughts on why the contamination noted in Figure 2 is most closely related to EAS?
- (7) For the rare variant analysis, could be nice to point out that for the individuals with SE ancestry, after the highest likelihood point in the tree in Figure S11.3, the next mostly likely branch is that leading to the IBS/TSI.
- (8) Pedigree – are there other possible pedigrees? How certain are we in the pedigree?
- (9) Figure 4 – Are those individuals of the same color within each section of the graph ordered in a certain way? Such as oldest to youngest? Or is it random?
- (10) New York is mentioned in the supplement, for doing shotgun sequencing (i.e., Supplement, p. 46), but it is not mentioned as a location in the Methods (main text, p. 15).
- (11) On p. 51 of the supplement, it says “We limited genotype calling to those sites with a genotyping Phred-scaled quality score or at least 45.” Does that mean that for one sample, some sites are genotyped while others use pseudo-haploid calls? It should be all one or the other.

(12) For genotyping, there can be a lot of uncertainty when using samples with coverage < 15, which is true of all the WGS individuals here. The genotyping was needed for the rare variant analysis, but were random alleles also called for them like for the other individuals? If not, is it possible to confirm other genetic results are robust when using random allele calling for these 10 individuals, and to add a sentence saying it is important to be cautious about how the genotyped data is used, and about what effect, if any, errors in genotyping might have had on the rare variant analysis (or any other analysis that depended on genotyping I might have missed).

(13) Because of the number of different datasets and analyses used here, a small table highlighting each major analysis, subsequent result/conclusion, and input data used might be useful for clarity.

(14) Add some new data from Olalde et al. 2018 "The Beaker phenomenon and the genomic transformation of northwest Europe"?

Reviewer #2 (Remarks to the Author):

Amorim et al present aDNA data from 63 samples from two cemeteries in Hungary and northern Italy. The main conceptual novelty is the approach by which they sequence all samples from the selected sites and analyse the DNA data comprehensively together with detailed archaeological and isotope data. In this way the authors are able to ask questions about social organization of the studied communities and also look at detailed ancestry and migratory patterns within the communities. One interesting result is that the Longobards in both locations had individuals with strikingly different ancestry profiles – including within kindreds. It is also intriguing that grave offerings correlate with genetic ancestry. This could mean that the Longobards were indeed a political union of different ethnical groups that kept cultural independence (united by cause or leadership?). Also strontium data from Szolad suggests the culturally and genetically different people may have moved to Szolad largely together (from the same place). The strontium results from Collegno are cool because they seem to document the arrival of Longobards and (social) mixture with locals (buried in same cemetery).

Overall the paper is clear in presenting the results and reads well. The analyses are generally thoughtfully executed the conclusions are supported by the data/analyses.

I recommend the publication of this paper with minor revisions

Page 6

“can be placed along the major northern and southern axis of modern European genetic variation” – consider rewording. There should be a more direct and descriptive way to describe the PC plot.

Figure 2

The figure – especially PCA – is difficult to follow. The PC plot could be larger. Maybe using shaded areas for the modern background instead of coloured dots would make the plot less busy? The thick and thin edges can be confusing. Maybe I don't see it but is the red circle with thick edge explained? Instead – what if you use coloured letters S and C (S1, C1, C2 for the kindreds) as symbols for plotting? Also – is it necessary to indicate Admixture proportions with symbol colour in the PC plot?

Page 7. I'm not sure the use of Population Assignment Analysis makes 100% sense. At the very least this paragraph should start by explaining why this is needed and what novelty it brings. Assigning aDNA to modern countries does not seem to add much to what is already evident from PCA. Also “See Figure 1 for color key” in Figure S10.1 is unclear..

A general note to consider regarding the first section of the results - “Genetic ancestry...” – The authors present five different groups/types of analyses. Given the space restrictions each gets somewhat limited attention, while all of them are quite complimentary. Maybe it would be better to concentrate on a few analyses in more detail in the main text and refer to others in supplement?

Reviewer #3 (Remarks to the Author):

I have been asked to review this paper as an historian, not as a biological scientist; so my comments are necessarily not about the science (whose methodology and accuracy I cannot judge), but exclusively about the historical context and the current historical debate within which this research sits.

Unsurprisingly the three historians involved in this project (La Rocca, Pohl and Geary) have done an excellent job of explaining the debates which have motivated this work - debates around the reality (or not) of migration, and around the coincidence (or not) of biological and cultural ethnicity. This is unsurprising since they are all experts in this field. Furthermore, although all three probably hoped they would find less coincidence between biological groupings and cultural groupings (as displayed by grave-type and grave-goods), and less evidence of an invading elite, they have very fairly presented the results and their implications; though I do note that they present their results phrased in the ultracautious negative: 'Thus our results cannot reject the migration, the route, and the settlement of "the Longobards" described in historical texts.'; rather than stating that they appear to support the traditional view of the migration, route and settlement of the Longobards.

This research is unquestionably novel and important; it will be widely studied and cited, not just with regard to the Longobards in Italy, but also in relation to other 'barbarian' peoples; and one can hope that it will stimulate other systematic work on cemeteries of this period. This is genuinely new data in a field where most research involves kicking around a few tired texts.

I very strongly recommend publication (assuming, of course, that the science is as good as it seems to be).

I also very much hope the authors will produce an article aimed at an audience of historians and archaeologists, with - for instance - illustration of the grave-goods and their connection to the genetics.

Response to Reviewer's Comments

Amorim et al. : Understanding 6th-Century Barbarian Social Organization and Migration through Paleogenomics

Corresponding Authors:

Johannes Krause,
David Caramelli,
Patrick J. Geary,
Krishna R. Veeramah

We thank all three reviewers for their comments. We have addressed each point by point, and hope they meet your satisfaction. We believe these changes have considerably improved the quality and clarity of the manuscript. Please see below our comments highlighted in bold.

Reviewer #1 (Remarks to the Author):

In Amorim et al., they study the genomic characteristics of individuals in two medieval cemeteries from w. Hungary (Szolad) and n. Italy (Collegno), archaeologically associated with the Longobard barbarian tribe. They wanted to test for the existence of a long distance migration into northern Italy by these people and to examine the genetic structure of these cemetery communities. They studied the genetic ancestry of as many interred individuals as possible and further studied the material culture, mortuary practices and isotope data associated with the remains. At both cemeteries, there were individuals of more 'northern European' (NE) ancestry and individuals of more 'southern European' (SE) ancestry, as well as identifiable family groups, with one predominant family group in each. The ones belonging to the large family group tended to be associated with NE ancestry. Using strontium isotope analysis, they showed that at Szolad, adults associated with both ancestries show diverse non-local ranges, while at Collegno older generations within the kin group had non-local strontium signatures, while more recent generations and those associated with southern European ancestry exhibit local strontium signatures.

The results of this paper are very interesting, especially the pedigree analysis and correlations in genetic and material/isotopic data, but it could benefit both from some reorganizing, additional genetic analyses, and a better discussion of how to interpret the analyses regarding genetic ancestry. Some major comments:

(1) Restructure introduction to address questions you actually answer. Current proposed questions (p. 3, bottom) are not answerable as they address all barbarian tribes, not just the Longobard, and even in the Longobard paragraph (p. 4), focus is on the long distance migration, which is not sufficiently discussed (perhaps because not enough evidence?). Move more quickly

to sampled cemeteries and specific questions addressing them – i.e., genetically who they are, internal genetic patterns to arrive at an understanding of kinship patterns, connection between genetics and grave good/elaborateness distribution, isotopic analysis and correlation to genetics/family. Perhaps a longer discussion on implications for understanding barbarian tribes can be included at the end, but that is not what this study addresses.

Authors' response: Following the reviewer's suggestions, we have re-organized and edited the introduction. We now omit from the first paragraph of the introduction the questions we did not address and instead focused on the ones we actually examine with our data (Pg 3, para 2). We also now only include two sentences that focus solely on the migration of the Lombards (which is necessary to justify our sampling of Szólád and Collegno), while the remaining parts of the introduction are focused on the questions of social organization (though sometimes this and migration are linked, such as for explaining the concept of *fara*) (Pg 3, para 3). However, we have still kept some of the more general historical and sociological description and discussion, as this was explicitly praised by reviewer 3 and feel it is important for the interdisciplinary audience this paper is aimed at.

(2) The bulk of the genetic section (p. 6-7) reiterates the same conclusion of northern-southern ancestry and takes up a large part of the paper. However, the discussion doesn't then dive into an exploration of whether long distance migration is the best explanation, and in fact back pedals with the sentence "However, whether the association between genetic ancestry and material culture reflects specific peoples mentioned in historical texts (i.e. Longobards) or stemmed from a deeper/long-term descent (of mixed barbarian ancestries) is as yet unclear (p. 11, bottom)". The rest of the discussion, however, seems to imply long distance migration was likely, particularly in Collegno (p. 13). And, what does it mean that in Szolad, we also see individuals of predominantly SE ancestry? You suggest they brought many foreign individuals to that area, but does that connect to any historic text and does it weaken conclusions for Collegno? I think this section is better served by having a shorter summary of the consistent genetic results and then detailing the different scenarios that could give rise to such genetic patterns (and what scenarios cannot). Then, it seems the non-genetic analyses, particularly the isotope one, can be used to help favor one hypothesis over another.

Authors' response: Regarding "*The bulk of the genetic section (p. 6-7) reiterates the same conclusion of northern-southern ancestry and takes up a large part of the paper. However, the discussion doesn't then dive into an exploration of whether long distance migration is the best explanation*", we note that determining the genetic ancestry of our samples is necessary in our study not just for looking at long-range migration, but is also vital for understanding the genetic structure within the cemeteries and how this is connected to the distribution of material culture in the subsequent sections (particularly how central/northern ancestry is associated with grave goods and southern ancestry is not). This finding in particular has been of great interest to people interested in the social structure of these populations. While for aDNA standards our samples are very well characterized genomically, the coverage is still low enough such that comparing all our

Szolad and Collegno individuals together is somewhat underpowered. As such, these insights about cemetery-wide structure are more powerful when we use the modern samples as a scaffold for inferring ancestry.

The second issue the reviewer raises, as far as we understand it, is about whether long-distance migration is the best explanation for the association between genetic ancestry and material culture, and points to the sentence “*whether the association between genetic ancestry and material culture reflects specific peoples mentioned in historical texts (i.e. Longobards) or stemmed from a deeper/long-term descent (of mixed barbarian ancestries) is as yet unclear*”. However, this particular sentence is not actually addressing the issue of migration of the Lombards. Instead, it is discussing the deeper issue of the correspondence between biological and cultural inheritance (note it is connected to the previous sentence where we suggest that “*the concept of long-term shared common descent in shaping social identity, as reflected in the material culture, may have had an actual biological basis in these two cemeteries*”. This emerges from the debate amongst scholars studying this period of whether barbarians mentioned in historical texts were as ethnically and genetically homogenous as described in the first 19th century studies. We note that reviewer 3 praised this aspect of our study, and thus think it is important to keep in the paper.

With regard to “*And, what does it mean that in Szolad, we also see individuals of predominantly SE ancestry? You suggest they brought many foreign individuals to that area, but does that connect to any historic text and does it weaken conclusions for Collegno?*” The southern ancestry individuals in Hungary do not seem to be local to the region based on isotope analysis and we speculate that they were foreigners brought to that area, as they also have a distinct material culture (or lack of it, see revised main text Pg 12, para 3 and Supplementary Note 15 for more discussion). Whether these individuals were slaves, servants, lower-caste members of the community, or something else, but certainly, given that Pannonia was part of the former Roman Empire, it is well known that under Roman rule, some soldiers from Italy settled and were granted land in Hungary after their discharge. Thus it would not be unusual that individuals from southern Europe may have ended up further north in Hungary. In contrast we see a different pattern to Collegno where SE individuals were all locals based on isotope analysis but first generation NE individuals were migrants. In addition, based on the Bronze Age data, we might expect Hungary to possess individuals that look most like modern southern Europeans, but also some with modern central/northern ancestry, while we would not expect individuals of this mix in Northern Italy (see below). This is now stated in the revised main text (Pg 13, para 1). Thus we do not believe the presence of SE ancestry in both cemeteries weakens the case of migration into Collegno.

We appreciate the opinion of the reviewer on how we should structure the discussion on long-distance migration and to address her/his concerns in this regard we have reworked

the section so the genetic data is summarized first, and then potential demographic scenarios are discussed in light of the isotope data (Pg 12, para 5 and pg 13, para 1).

(3) A large assumption here is that present-day distributions do reflect what the distribution should look like during this time period. I am less convinced without ancient data to help confirm this. For instance, based on the genetic analysis, I could see an argument that northern Italy was predominantly of NE ancestry at the time, and more and more people of SE ancestry entered that area, leading to much more SE ancestry today. I don't necessarily believe that is true, but I don't think that demographic model has been eliminated. A deeper look into the validity of this assumption is worth having, and if possible, using more published ancient data to determine whether the assumption is valid would be useful.

(4) For the PCA analysis, the supplemental text says: "It is also noteworthy that samples from Szólád do not appear to be clustered next to Neolithic, Bronze Age or modern samples from Hungary. This lack of overlap with the Bronze Age is made more apparent by performing this analysis only with Bronze age, medieval and modern samples" (p. 56). However, at least in Figure S7.4, if I interpret it correctly, the BB (Bell Beaker), EBr (Eastern European Bronze Age), HBr (Hungarian Bronze Age), NBr (northern European Bronze Age) at the least all seem to overlap with both some Collegno and Szolad samples. Thus, there are overlaps with Bronze Age samples, and I think this helps your argument of more central/northern European ancestry in some samples (from both cemeteries). I think this is important to describe more clearly, both in the supplement and main text.

(5) In the Mathieson et al. 2015 paper, there are 2-3 samples in their Figure 1a that are found in northern Italy. I'm not sure who those individuals are, but I think it is worth highlighting them, as if they have northern/central European ancestry and are older, then it would suggest there was already such ancestry in this region previously, making it more difficult to argue that the presence of northern ancestry is related to a Longobard migration.

Authors' response: As they are connected, we respond to 3,4 and 5 together. Since our paper was in review, three relevant papers have been released, Mathieson et al. 2018 Nature, Olalde et al. 2018 Nature and Damgaard et al. 2018. Along with the original Mathieson et al. 2015 data, we have included additional Bronze Age European data from the first two and Western Eurasian migration period and East Hungarian Scythian individuals from the last.

We note that the 3 Italian individuals (along with the Iceman) from Mathieson et al. 2015 were classified as CEM (Central European Early and Middle Neolithic). Unsurprisingly, given their age, they cluster towards early farmers, near modern Sardinians. We now indicate them in our PCA (Fig 2B) as "ITn". The study of Olalde et al. introduced 6 Bell Beaker individuals from Italy (3 from Northern Italy, 3 from Sicily). All cluster towards southern and western of modern European and their median values are intermediate with the CEM individuals (though they are quite spread, one overlapping modern southern Europeans and one overlapping modern western Europeans). These are indicated in our new PCA in Fig 2B as ITb. There is no evidence of major central/northern

European ancestry in these individuals. We also note that three Individuals from Bronze Age Switzerland are closer to the central/northern than southern cluster (though towards the “southern” end of the former) (Supplementary Text pg 34, para 2, Supplementary Figure 79), which supports previous work (Petkova et al. 2015 Nat Gen) of the Alps being a major long-term barrier to gene flow between central Europe and northern Italy.

Other than modern Europeans, as the Bronze Age is temporally the closest era to the Migration Period for which there exist a large number of ancient genomes, we examined how the Bronze Age related to the Migration Period and modern Europe more generally. We have conducted a more in depth PCA as well as a series of D-statistics analyses to show that there is a rough similarity between the genetic structure modern day and Bronze Age Europe in terms of northern and southern ancestry, and that our samples with high CEU-GBR ancestry would have been unusual in Bronze Age Italy, and even Hungary when compared to northern and central Europe (Supp Notes 7 and 11).

In addition, Damgaard et al. introduced a small number of contemporary samples to our migration period samples. In addition to the 7 Anglo-Saxon-era individuals from Schiffels et al. the 2 Bavarian individuals from Veeramah et al. 2018 PNAS, 5 Alan individuals sampled from the Caucasus (CAm) and 1 individual from Lithuania (NEM) all cluster close their modern counterparts. The only exception to this general concordance is a migration period sample from Slovakia (SKm), which is found amongst our more northern European cluster. However, it should be noted that this sample may be unusual, as it noted to be a double-chambered and elaborate chieftain grave that shows many similarities with elite Germanic chamber graves from the Late Roman Period. Thus, though the sample size is currently low, in general other ancient West Eurasian samples from this period appear to show a fairly consistent relationship with modern individuals from a similar area.

We now discuss these findings in the revised main text in both the results (Pg 7, para 3) and the discussion (Pg 12, para 5 and pg 13, para 1) as well as in the Supplementary Text and hope it meets to the satisfaction of the reviewer.

(6) Most Admixture analyses are unsupervised, including samples from potential reference populations and projecting samples with few data (like most ancient individuals). Why was an unsupervised analysis, estimating the CVV to determine the best K groups, not done? I would want to see an unsupervised analysis to make sure unknown biases are not being introduced with the supervised analysis.

Authors' response: A supervised analysis was initially performed in order to mitigate against the low population genetic differentiation amongst modern European populations by more precisely defining putative ancestral allele frequencies. As requested, we have now performed CV with unsupervised analysis for both our 1000 Genomes and prehistorical-based analyses. For the former, as anticipated it is difficult to form clear

clusters amongst modern European populations (primarily because such a model is probably mis-specified for this kind of isolation-by-distance data). However, despite this, the same general trend is seen, with our ancient individuals again showing ancestry that is almost exclusively found in modern European populations, with the same northern and southern-related ancestry associated with individuals from the supervised analysis. For the latter (analysing the samples alongside WHG, EEF and SA prehistorical samples) the results are nearly identical for the supervised and unsupervised analyses. These new results have been added to Supplementary Note 8 and are mentioned in the revised main text (Pg 6, para 3, “*An unsupervised analysis on a set of unrelated samples from Szólád and Collegno demonstrated a similar CEU+GBR vs TSI- like structure (Supplementary Figures 44-46)*” and Pg 7, para 2, “*Both PCA and supervised and unsupervised ADMIXTURE analysis ((Supplementary Figures 65-72))*”)

(7) D-statistics: The major analyses assigning genetic ancestry are all based on more exploratory analyses. I believe there is enough data here that you can do D-statistics (and f3-statistics) to more directly test which connections these ancient individuals share to each other and other European populations. Testing $D(\text{ancient}, X; \text{ancient}, \text{Mbuti/Chimp})$ would help establish whether these ancient individuals form a clade with each other relative to other populations/individuals (X), and what types of admixture is characteristic of this population. Adding $D(\text{ancient}, \text{ancient}; X, \text{Mbuti/Chimp})$ where X are southern Italians/Spanish or northern Hungarians/central Europe populations/individuals would be the best confirmation of conclusions of asymmetric NE/SE ancestry made in ADMIXTURE/PCA.

Authors' response: We initially avoided D and f-statistic analysis in our original draft, as the close genetic similarity amongst European populations generally makes this approach underpowered and extensive gene flow (as evidenced by the strong isolation-by-distance pattern observed for modern European) may violate the underlying model. However, as requested by the reviewer, we have performed an extensive D-statistic analysis involving almost 500,000 individual tests in a framework that tries to identify if any two unrelated samples from Szolad and Collegno form a clade versus any modern population. As expected, this is generally difficult, and in most cases the best we can say is that we cannot statistically rule out two samples forming a clade against other modern populations. However, there are still clear patterns that correlate with whether the two test samples are of both the same ancestry (i.e. central/northern v central/northern and southern v southern) or opposing ancestries (i.e NE v SE). In addition, we did identify some pairs of samples of northern ancestry that could be considered significant clades, including samples from different cemeteries. We have provided an extensive write up of this in the Supplementary Material (Supplementary Note 11) and also discuss the results in the revised main text (Pg 7, para 1) as follows: “*A series of D-statistics analyses of the form $D(\text{ancient}, \text{ancient}, \text{modern_reference}, \text{outgroup})$ confirms the close relationship amongst our ancient samples assigned as being of primarily central/northern ancestry from both Szólád and Collegno (there are even four Szólád-Collegno pairs that appear to form significant clade compared to all*

other modern populations), while those of southern ancestry show more potential structure (Supplementary Note 11, Supplementary Figures 51-60, Supplementary Table 7)."

Minor comments:

(1) In the main text, especially the abstract, make sure it is clear what terms such as barbarian, medieval, Pannonia, refer to for the average reader.

Authors' response: To address the reviewer's suggestion, we added some details to these terms in the Introduction (Pg 3, para 1 - "Early Middle Ages (i.e. 3rd to 10th centuries CE)"; Pg 3, para 1 - "various western and eastern barbarian/non-Roman peoples"; Pg 4, para 1 - *Roman province of Pannonia (what is now Western Hungary and Lower Austria)*). We were not able to add these to the abstract, since there is a strict word limit for this.

(2) On p. 5, please list mean and range of radiocarbon dates directly in main text.

Authors' response: Our method for carbon dating yields a range for absolute calendar ages for several different individuals, but unfortunately not a point estimate. Ranges of radiocarbon dates are now added to the main text (Pg 4, para 3, "*(2-sigma range of 412-604 CE, Supplementary Table 1)*").

(3) The PCA figures are very hard to read. I would suggest making modern individuals gray-scaled, and using triangles instead of stars to reduce the number of edges. I would also consider not using the border color/thickness – it's a subtle detail that is very hard to distinguish.

Authors' response: We have remade our PCA plots, increasing their size and also splitting the modern and ancient reference samples onto two different panels (this became more important as we added new ancient reference samples). In addition we have changed the stars to triangles and removed the border color/thickness as requested. We grey-scaled the modern populations in the 2nd PCA where the focus is the ancient reference samples. We do keep the colours of the modern populations in the first PCA however, as we believe this provides important contextual information about the size of the spread of the different modern regions. We hope the new version is easier to read.

(4) Are the results for the Bronze Age samples relative to present-day populations consistent with previous analyses? Is there any ascertainment bias in using the Mathieson et al. data?

Authors' response: The placement of the previously published ancient samples on the PCA appears to be consistent with their associated previous studies. The orientation of samples along the PC1 and PC2 axis differs as we use of a different modern reference dataset, but the relative placement does not appear to show any major differences with published works.

With regard to ascertainment bias, we assume the reviewer is referring to the use of the 1240K SNP capture. Obviously some ascertainment bias is always possible simply by using previously known SNPs, however with this capture having been used in numerous other high profile studies of European prehistory. In addition, we note the strong concordance between the SNP-based analysis and the rare variant analysis (which does not use the ascertained SNP capture array), suggesting any bias is minimal. This note has been added to the revised main text (pg 6, para 2 “*This north/south axis of genetic variation is also observed when examining only our ancient samples, demonstrating that our results are not a bias introduced due to the reliance on modern reference populations or close kinship*”). In addition, as shown by Patterson et al. Genetics. the use of an outgroup should mitigate any ascertainment bias that might occur in our new D-statistic analyses.

If the reviewer is referring to ascertainment bias introduced because of the choice of ancient sample selection, we note that our PCA are based on individual plots of an ancient sample against the modern reference datasets, followed by procrustes transformation, so no ancient reference sample will have any effect on any other ancient reference sample (as they are never directly analyzed together). They are analyzed together in the ADMIXTURE analysis, but the results are highly concordant with the PCA analysis, suggesting limited if any ascertainment bias.

(5) Published ancient data cites Mathieson et al. (2015), but I think most of their data came from a previously published dataset in Lazaridis et al. 2014 and Allentoft et al. 2015.

Authors’ response: The data in Mathieson et al. actually derives from a number of different sources. We do not provided references for all of them as this list would be very long, but we do now acknowledge the multiple origins in Supplementary Note 6 (Pg 29, para 3), as well as providing the original source for each sample in a new Supplementary Table 3.

(6) Any thoughts on why the contamination noted in Figure 2 is most closely related to EAS?

Authors’ response: We hypothesize that East Asian contamination is likely a result of plastic wares produced in Asia that were utilized in DNA extraction, as we have occasionally seen this in some of our previous negative controls. We have now noted this in the Supplementary text (Supplementary Note 7, Pg 31, para 2, “*this sample showed high levels of contamination (which we hypothesize is the result of plastic wares that were utilized in DNA extraction)*”).

(7) For the rare variant analysis, could be nice to point out that for the individuals with SE ancestry, after the highest likelihood point in the tree in Figure S11.3, the next mostly likely branch is that leading to the IBS/TSI.

Authors’ response: Unfortunately, other requested additions to the manuscript led to its length going over the 5,000 word limit, so we do not have room to do this in the main

text. However, we do mention it in the supplementary text (Supplementary Note 12, Pg 59, para 1).

(8) Pedigree – are there other possible pedigrees? How certain are we in the pedigree?

Authors' response: Given the kinship results estimated with IcmIkin with four different assumptions for population allele frequencies (see supplementary section on “Biological kinship inference”), there are no other possible pedigrees for kindreds SZ1, CL1, CL2 (now Fig 4). We were not able to choose a single pedigree for the other four smaller kindreds. Both the pedigrees depicted on Fig 4 and the lack of a single best choice to depict the pedigree of the other four kindreds were confirmed manually by two investigators in this work (CEGA and KRV) independently, and also by running PRIMUS, a software designed to infer pedigrees from genome-wide estimates of IBD (see reference Staples et al 2014, Doi:10.1016/j.ajhg.2014.10.005). In summary, we manually inferred pedigrees by starting from non-ambiguous relationships (e.g. parent and child) and then extended the family tree to the more ambiguous cases. When a parent/-child or grandparent-grandchild relationship was inferred, we defined the parent (or grandparent) as the individual who was the oldest at the moment of death, according to anthropological estimates. With one exception (discussed in detail in the supplements), the relationship between one pair of individuals was always corroborated by the relationship of each one of the individuals in this pair to a third individual. Because of the approach we took, we are largely certain about the pedigrees inferred for kindreds SZ1, CL1, CL2. For further details, please see Supplementary Note 13 on “Biological kinship inference”.

(9) Figure 4 – Are those individuals of the same color within each section of the graph ordered in a certain way? Such as oldest to youngest? Or is it random?

Authors' response: To ease readability of the figure (now Fig 5), we plotted the $^{87}\text{Sr}/^{86}\text{Sr}$ values within each ancestry group arranged in ascending order. We added a note to that matter to the Figure's caption and corrected the order of two samples that were mistakenly not following this pattern.

(10) New York is mentioned in the supplement, for doing shotgun sequencing (i.e., Supplement, p. 46), but it is not mentioned as a location in the Methods (main text, p. 15).

Authors' response: a mention to the NYGC was added to the revised main text.

(11) On p. 51 of the supplement, it says “We limited genotype calling to those sites with a genotyping Phred-scaled quality score or at least 45.” Does that mean that for one sample, some sites are genotyped while others use pseudo-haploid calls? It should be all one or the other.

Authors' response: What we meant to state was that to call a pseudo-haploid genotype, the phred-scale quality of the call had to have a phred-scaled variant quality of at least 45, we did not conduct a mix of pseudo-haploid and diploid genotype calls for any

particular analysis (though for IcMLkin used genotype likelihoods, and for the rarecoal analysis we also used diploid calls for all 10 whole genomes). We apologize for the confusion, we have changed the sentence to “We limited *haploid* genotype calling to those sites with a Phred-scaled quality score of at least 45” (Supplementary Text, pg 26, para 3).

(12) For genotyping, there can be a lot of uncertainty when using samples with coverage < 15, which is true of all the WGS individuals here. The genotyping was needed for the rare variant analysis, but were random alleles also called for them like for the other individuals? If not, is it possible to confirm other genetic results are robust when using random allele calling for these 10 individuals, and to add a sentence saying it is important to be cautionary about how the genotyped data is used, and about what effect, if any, errors in genotyping might have had on the rare variant analysis (or any other analysis that depended on genotyping I might have missed).

Authors’ response: As noted in the original Supplementary Text, we already have performed the rare variant analyses using both pseudo-haploid and diploid genotype calls:

“We then called both pseudo-haploid and diploid genotypes as described in Supplementary Note 5 for our medieval whole genomes in the same regions, restricting to sites with 7x depth or greater.”

We also included this statement at the end of the original supplementary text.

“Using diploid calls (still ignoring singletons) appeared to improve our ability to place samples on the southern portion of the tree (Figure S11.4), with SZ36 placed on the shallow TSI branch, and the likelihood distribution being much sharper across the tree space, though we must be cautious in interpreting these results due to the variable WGS coverage that may result in overconfident placement due to unaccounted diploid calling error.”

We hope the original text satisfies the reviewer’s concerns. In addition we have added a note the Methods section of the revised main text (Pg 15, para 4, “Analysis was performed using both pseudo-haploid calls and diploid genotypes”).

(13) Because of the number of different datasets and analyses used here, a small table highlighting each major analysis, subsequent result/conclusion, and input data used might be useful for clarity.

Authors’ response: In this new version of the manuscript, we include a supplementary table (Supplementary Table 19) listing the main results generated with genetic data with details about input data used (reference datasets and how genetic data was treated). In order to keep it concise, we did not include a summary of corresponding conclusions, but we indicate which figures result from the corresponding analysis and point the reader to the the particular supplement where details about the analysis and some discussion/conclusions can be found.

(14) Add some new data from Olalde et al. 2018 “The Beaker phenomenon and the genomic transformation of northwest Europe”?

Authors’ response: Please see response to 3, 4 and 5.

Reviewer #2 (Remarks to the Author):

Amorim et al present aDNA data from 63 samples from two cemeteries in Hungary and northern Italy. The main conceptual novelty is the approach by which they sequence all samples from the selected sites and analyse the DNA data comprehensively together with detailed archaeological and isotope data. In this way the authors are able to ask questions about social organization of the studied communities and also look at detailed ancestry and migratory patterns within the communities. One interesting result is that the Longobards in both locations had individuals with strikingly different ancestry profiles – including within kindreds. It is also intriguing that grave offerings correlate with genetic ancestry. This could mean that the Longobards were indeed a political union of different ethnical groups that kept cultural independence (united by cause or leadership?). Also strontium data from Szolad suggests the culturally and genetically different people may have moved to Szolad largely together (from the same place). The strontium results from Collegno are cool because they seem to document the arrival of Longobards and (social) mixture with locals (buried in same cemetery).

Overall the paper is clear in presenting the results and reads well. The analyses are generally thoughtfully executed the conclusions are supported by the data/analyses.

I recommend the publication of this paper with minor revisions

Page 6

“can be placed along the major northern and southern axis of modern European genetic variation” – consider rewording. There should be a more direct and descriptive way to describe the PC plot.

Authors’ response: Following the reviewer’s suggestions, we rephrased the corresponding sentence in order to make it more descriptive of the PCA findings. The current sentence (Pg 6, para 2) reads as follows:

“However, they do not cluster with individuals from their respective modern countries of origin. Instead, samples from both cemeteries demonstrate a diverse distribution, with two broad clusters around modern northern and southern individuals, as well as individuals of intermediate ancestry. This north/south axis of genetic variation....”

Figure 2

The figure – especially PCA – is difficult to follow. The PC plot could be larger. Maybe using

shaded areas for the modern background instead of coloured dots would make the plot less busy? The thick and thin edges can be confusing. Maybe I don't see it but is the red circle with thick edge explained? Instead – what if you use coloured letters S and C (S1, C1, C2 for the kindreds) as symbols for plotting? Also – is it necessary to indicate Admixture proportions with symbol colour in the PC plot?

Authors' response: As also described in response to reviewer 1, we have remade our PCA plots, increasing their size and also splitting the modern and ancient reference samples onto two different panels (this became more important as we added new ancient reference samples). We experimented with shaded areas, but this actually became more confusing with a lot of overlap between shaded areas, as while there is a general pattern of IBD in Europe, there are still always a few individuals that are located fairly distance from other individuals from the same country of origin. Similarly, using letters for the kindreds also made the plot unreadable as often letters from the same kindred would overlap. We have now simply removed the kinship information from the main text plot, and instead highlighted them as a separate figure in the supplementary material (Supplementary Figure 88). However, we felt keeping the admixture proportion symbol colors was important for maintaining continuity between the other 4 main figures.

Page 7. I'm not sure the use of Population Assignment Analysis makes 100% sense. At the very least this paragraph should start by explaining why this is needed and what novelty it brings. Assigning aDNA to modern countries does not seem to add much to what is already evident from PCA. Also "See Figure 1 for color key" in Figure S10.1 is unclear.

Authors' response: We consider PCA as used in this context to be a purely exploratory tool, which will be influenced by sample size of the reference population, SNP choice (which is different for each ancient sample in this case because of coverage) and which PCs are examined, and provides no assessment of error, which is important given how similar modern Europeans even from different countries are to each other. Our justification for PAA provides an actual probability and measure of uncertainty of being related to a particular group of individuals that is relatively robust to these effects. This turns out to be important in understanding how well we can distinguish ancestry between central and northern Europeans. We have revised a short statement in the revised main text (Pg 6, para 4) to state that the test provides an estimate of uncertainty: *"Population Assignment Analysis (PAA) that estimates uncertainty in genetic ancestry assignment finds that individuals with high CEU+GBR ancestry are assigned to countries from all over modern central and northern and northwest Europe"*.

In addition, a new descriptive color scheme has been given for the legend of Figure S10.1 (now Supplementary Figure 50).

A general note to consider regarding the first section of the results - "Genetic ancestry..." – The authors present five different groups/types of analyses. Given the space restrictions each gets

somewhat limited attention, while all of them are quite complimentary. Maybe it would be better to concentrate on a few analyses in more detail in the main text and refer to others in supplement?

Authors' response: Given the changes requested by reviewer 1 to incorporate more ancient sample analysis, this section has been considerably altered, and as such we could not simply modify the original text as requested by reviewer 2 (though description of some analyses have now been considerably shortened). We also note that *Nature Communications* requires all supplementary text to be mentioned in the main text in some form.

Reviewer #3 (Remarks to the Author):

I have been asked to review this paper as an historian, not as a biological scientist; so my comments are necessarily not about the science (whose methodology and accuracy I cannot judge), but exclusively about the historical context and the current historical debate within which this research sits.

Unsurprisingly the three historians involved in this project (La Rocca, Pohl and Geary) have done an excellent job of explaining the debates which have motivated this work - debates around the reality (or not) of migration, and around the coincidence (or not) of biological and cultural ethnicity. This is unsurprising since they are all experts in this field. Furthermore, although all three probably hoped they would find less coincidence between biological groupings and cultural groupings (as displayed by grave-type and grave-goods), and less evidence of an invading elite, they have very fairly presented the results and their implications; though I do note that they present their results phrased in the ultracautious negative: 'Thus our results cannot reject the migration, the route, and the settlement of "the Longobards" described in historical texts.'; rather than stating that they appear to support the traditional view of the migration, route and settlement of the Longobards.

Authors' response: We believe being “ultracautious” at this stage is important until we can obtain further data to establish what the genetic background of Europe was immediately before the Migration Period. Only then can we truly be able to infer whether migrations have taken place, versus the patterns we observe actually reflecting the long term structure of the regions.

This research is unquestionably novel and important; it will be widely studied and cited, not just with regard to the Longobards in Italy, but also in relation to other 'barbarian' peoples; and one can hope that it will stimulate other systematic work on cemeteries of this period. This is genuinely new data in a field where most research involves kicking around a few tired texts.

I very strongly recommend publication (assuming, of course, that the science is as good as it seems to be).

I also very much hope the authors will produce an article aimed at an audience of historians and archaeologists, with - for instance - illustration of the grave-goods and their connection to the genetics.

Authors' response: We thank the reviewer for their positive comments. An article targeted towards historians and archaeologists is indeed being developed.

Reviewer #1 (Remarks to the Author):

Overall, I think the paper is much improved, primarily in organization and clarity, as well as having the extra D-statistic analysis, where when possible, the patterns observed are consistent with what was found using more descriptive methods. The figures are much clearer, especially the PCA plots. Aside from a few minor comments, I believe the revisions included clarify the major conclusions and confirm them with the extra analyses.

Minor comments:

There are quite a few small grammatical errors that need fixing. (double commas, double periods, typos)—I don't mention them all, but at least a few:

--in supplement, p. 54, S11.3, extra parantheses after HUb

—last line on p. 3, two commas

—fourth to last line on p. 3 - two ?? and also, should it be late-antiquity?

—middle paragraph on p. 5 - two periods

p.4 Does the quote need a citation?

p. 28 - Figure 5, Can you put in the figure legend what the triangle and connecting lines mean in the Szolad part of the figure? Also, because the black color is not an ancestry group, perhaps better to use different terminology in the last sentence.

Reviewer #2 (Remarks to the Author):

The authors have addressed all points I raised adequately, I have no further comments/concerns

Manuscript details:

Amorim et al.: Understanding 6th-Century Barbarian Social Organization and Migration through Paleogenomics

Corresponding Authors:

Johannes Krause,
David Caramelli,
Patrick J. Geary,
Krishna R. Veeramah

Response to Reviewer's Comments

Reviewer #1 (Remarks to the Author) - Minor comments:

There are quite a few small grammatical errors that need fixing. (double commas, double periods, typos)—I don't mention them all, but at least a few: - **We have performed additional error checking beyond the points below.**

--in supplement, p. 54, S11.3, extra parantheses after HUb - **We believe the reviewer may be referring to the older supplement. We cannot find the instance of this in the revised supplement.**

—last line on p. 3, two commas - **FIXED**

—fourth to last line on p. 3 - two ?? and also, should it be late-antiquity? - **FIXED, though “late antique” is the correct term.**

—middle paragraph on p. 5 - two periods - **FIXED**

p.4 Does the quote need a citation? - **We have added a citation for this quote.**

p. 28 - Figure 5, Can you put in the figure legend what the triangle and connecting lines mean in the Szolad part of the figure? Also, because the black color is not an ancestry group, perhaps better to use different terminology in the last sentence - **We have changed the figure to get rid of the triangles, and use only circles. The circle and triangle representation was supposed to represent sampling of specific molars, M2 and M3, but we have checked and the original study (Alt et al.) actually used multiple molar types. We figure legend has also been adjusted to reflect this.**